# Whole genome genetic variation and linkage disequilibrium in a diverse collection of *Listeria monocytogenes* isolates

**Swarnali Louha**[1]*, **Richard J. Meinersmann**[2], **Travis C. Glenn**[1,3]

**1** Institute of Bioinformatics, University of Georgia, Athens, GA, United States of America, **2** USDA Agricultural Research Service, U.S. National Poultry Research Center, Athens, GA, United States of America, **3** Department of Environmental Health Science, University of Georgia, Athens, GA, United States of America

* sl50708@uga.edu

**Data Availability Statement:** All relevant data are within the manuscript and its Supporting Information files. All data used in this study have

## Abstract

We performed whole-genome multi-locus sequence typing for 2554 genes in a large and heterogenous panel of 180 *Listeria monocytogenes* strains having diverse geographical and temporal origins. The subtyping data was used for characterizing genetic variation and evaluating patterns of linkage disequilibrium in the pan-genome of *L. monocytogenes*. Our analysis revealed the presence of strong linkage disequilibrium in *L. monocytogenes*, with ~99% of genes showing significant non-random associations with a large majority of other genes in the genome. Twenty-seven loci having lower levels of association with other genes were considered to be potential "hot spots" for horizontal gene transfer (i.e., recombination via conjugation, transduction, and/or transformation). The patterns of linkage disequilibrium in *L. monocytogenes* suggest limited exchange of foreign genetic material in the genome and can be used as a tool for identifying new recombinant strains. This can help understand processes contributing to the diversification and evolution of this pathogenic bacteria, thereby facilitating development of effective control measures.

## Introduction

The bacterial genome is a dynamic structure. Characterizing patterns of genomic variation in bacterial pathogens can provide insights into the forces shaping their biology and evolutionary history [1]. Homologous recombination is an important driver of evolution and increases the adaptive potential of bacteria by allowing variation to be tested across multiple genomic backgrounds [2]. Recombination is mediated by three mechanisms; transformation, transduction, and conjugation, and the availability and efficacy of these mechanisms and their biological consequences play a major role in determining the frequency of recombination in a bacterial population [1, 3]. Recombination is variably distributed in bacterial genomes, with some sites in the genome recombining at a higher or lower frequency than the genomic average, known as hot spots and cold spots respectively [4]. Evidence for recombination and its effect on genomic variation can be obtained by detecting patterns of non-random association of genotypes at different loci within a given population, termed as linkage disequilibrium [1, 3]. Various

been uploaded to GenBank and the Accession numbers have been recorded in S1 File.

**Funding:** This research was supported by funding from USDA Agricultural Research Service Project Number 6040-32000-009-00-D. The funders helped in data collection and preparation of the manuscript.

**Competing interests:** The authors have declared that no competing interests exist.

methods for detecting linkage disequilibrium have been used to study the extent of genetic recombination shaping the population structures of several bacterial species [1, 5–7].

*Listeria monocytogenes*, known for causing life-threatening infections in animals and human populations at risk, is one of the bacterial species having the lowest rate of homologous recombination. Genetic diversity in this species is mainly driven by the accumulation of mutations over time, with alleles five times more likely to change by mutation than by recombination [8]. *L. monocytogenes* is generally considered to have a clonal genetic structure [9, 10]. The population structure of this bacteria consists of 4 evolutionary lineages (I, II, III and IV) and recombination has been observed between isolates of different lineages; suggesting that although recombination is rare in *L. monocytogenes*, this species is not completely clonal [8, 11, 12]. Interestingly, homologous recombination is not equally frequent among isolates of different lineages, with lineages II, III and IV showing higher rates of recombination and lower degree of sequence similarity than lineage I [11, 13–15].

Whole-genome sequencing studies have shown that *L. monocytogenes* genomes are highly syntenic in their gene content and organization, with a majority of gene-scale differences occurring in the accessory genome and accumulated in a few hypervariable hotspots, prophages, transposons, scattered unique genes and genetic islands encoding proteins of unknown functions [14, 16–19]. Several other studies have detected evidence of recombination using a few genes [8, 11, 20] and indicated the presence of significant linkage disequilibrium in *L. monocytogenes* [21, 22]. However, these studies used a limited number of *L. monocytogenes* isolates and evaluated recombination present in a small fraction of the genome, mostly made up of house-keeping genes, which are assumed to be under negative selection and less subject to homologous recombination.

Prior to the advent of next-generation sequencing technologies, multi locus enzyme electrophoresis (MLEE), was used for generating large data sets for the statistical analysis of bacterial populations. MLEE differentiates organisms by assessing the relative electrophoretic mobilities of intracellular enzymes and indexes allelic variation in multiple chromosomal genes [23]. MLEE has been successfully used for studying the extent of linkage disequilibrium in a variety of bacterial species [5, 9, 24]. With the easy and cheap availability of sequencing data in the last decade, MLEE has been replaced with an analogous technique called MLST (multi locus sequence typing) for subtyping bacterial genomes [22, 25]. We recently provided an approach that can generate whole-genome MLST (wgMLST) based characterization of *L. monocytogenes* isolates from whole-genome sequencing data [26]. In this study, we use this wgMLST-based approach for characterizing genomic variation and assessing genome-wide patterns of linkage disequilibrium in a large collection of *L. monocytogenes* isolates obtained from diverse ecological niches.

## Materials and methods

### *Listeria monocytogenes* isolate selection

We selected a large and diverse panel of 180 *L. monocytogenes* isolates collected from different ecological communities (S1 File). This set included (i) 20 isolates each from food, food contact surfaces (FCS), manure, milk, clinical cases, soil, and ready-to-eat (RTE) products, for which whole-genome sequencing data was obtained from the NCBI Pathogen Detection database and, (ii) 20 isolates from water and sediment samples in the South Fork Broad River watershed located in Northeast Georgia and 20 isolates from effluents from poultry processing plants (EFPP), for which whole-genome sequencing data was provided by the USDA and FSIS [26].

## Whole-genome multi-locus sequence typing (wgMLST)

Whole-genome sequencing data for the 180 *L. monocytogenes* isolates were processed using Haplo-ST (S1 Fig, [26]) for allelic profiling of 2554 genes per isolate. Haplo-ST first cleaned raw Illumina whole-genome sequencing reads obtained as previously described (S1 File) using the FASTX-Toolkit [27]. Next, reads were trimmed to remove all bases with a Phred quality score of < 20 from both ends and filtered such that 90% of bases in the clean reads had a quality of at least 20. After trimming and filtering, all remaining reads with lengths of < 50 bp were filtered out. Next, Haplo-ST used YASRA [28] to assemble the cleaned reads into allele sequences and provided wgMLST profiles to the assembled allele sequences with BIGSdb-*Lm* (available at http://bigsdb.pasteur.fr/listeria).

## Analysis of linkage disequilibrium

First, the raw wgMLST profiles were filtered to remove paralogous loci and genes were ordered according to their genomic position in the *L. monocytogenes* reference strain EGD-e (NCBI Accession number NC_003210.1). Next, new alleles not defined in the BIGSdb-*Lm* database and reported as 'closest matches' to existing alleles in BIGSdb-*Lm* were assigned custom allele ID's with in-house Python scripts. The wgMLST profiles were further filtered to retain loci with < 5% missing data. The remaining loci were used to evaluate linkage disequilibrium (LD) between all pairs of loci with Arlequin v3.5.2 [29]. LD tests for the presence of significant statistical association between pairs of loci and is based on an exact test. The test procedure is analogous to Fisher's exact test on a two-by-two contingency table but extended to a contingency table of arbitrary size [30]. For each pair of loci, first a contingency table is constructed. The $k_1$ x $k_2$ entries of this table are the observed haplotype frequencies, with $k_1$ and $k_2$ being the number of alleles at locus 1 and locus 2, respectively. The LD test consists in obtaining the probability of finding a table with the same marginal totals and which has a probability equal or less than that of the observed contingency table. Instead of enumerating all possible contingency tables, a Markov chain is used to explore the space of all possible tables. To start from a random initial position in the Markov chain, the chain is explored for a pre-defined number of steps (the dememorization phase), such as to allow the Markov chain to forget its initial phase and make it independent from its starting point. The *P*-value of the test is then taken as the proportion of the visited tables having a probability smaller or equal to the observed contingency table. In our analysis, we used 100,000 steps of Markov chain to test the *P*-value of the LD test and 10,000 dememorization steps to reach a random initial position on the Markov chain. The significance level of the LD test was set at a *P*-value of 0.05.

## Assessment of genetic diversity

Genetic diversity between *L. monocytogenes* isolates collected from the different ecological niches listed as the isolate sources (S1 File) was computed with pairwise $F_{ST}$'s in Arlequin. $F_{ST}$ measures the proportion of the variance in allele frequencies attributable to variation between populations [31] and has a history of being used as a measure of the level of differentiation between populations in population genetics [32, 33]. Fifty thousand permutations were used to test the significance of the genetic distances at a significance level of 0.05.

The AMOVA procedure in Arlequin was used to compute the pairwise differences in allelic content between isolate wgMLST profiles as a matrix of Euclidean squared distances. This distance matrix was used to compute a minimum spanning tree (MST) between all isolates. The MST was visualized and annotated with iTOL v3 [34]. For better visualization, the MST was converted to circular format and annotations for the source of isolates were displayed in outer external rings.

## Results

We performed whole-genome multi locus sequence typing for 180 *L. monocytogenes* isolates obtained from 9 different source populations. For each isolate, allele sequences were assembled for 2554 genes and provided allele ID's based on the unified nomenclature available in the BIGSdb-*Lm* database (S2 File). This dataset was filtered to remove 133 paralogous loci identified by Haplo-ST and all loci with > 5% missing data (alleles not assigned ID's by Haplo-ST), and the remaining 2233 loci (S3 File) were ordered according to their position in the *L. monocytogenes* reference genome EGD-e. Fig 1 shows the minimum spanning tree of the 180 isolates inferred from allelic differences in the wgMLST profiles. Two results are apparent. First, we see a long branch (red) containing a majority of isolates obtained from soil and manure clustered together, which suggests the origin of these strains from a common ancestor. Interestingly, three clinical strains (SRR1030275, SRR974870, SRR974873) are also found in this

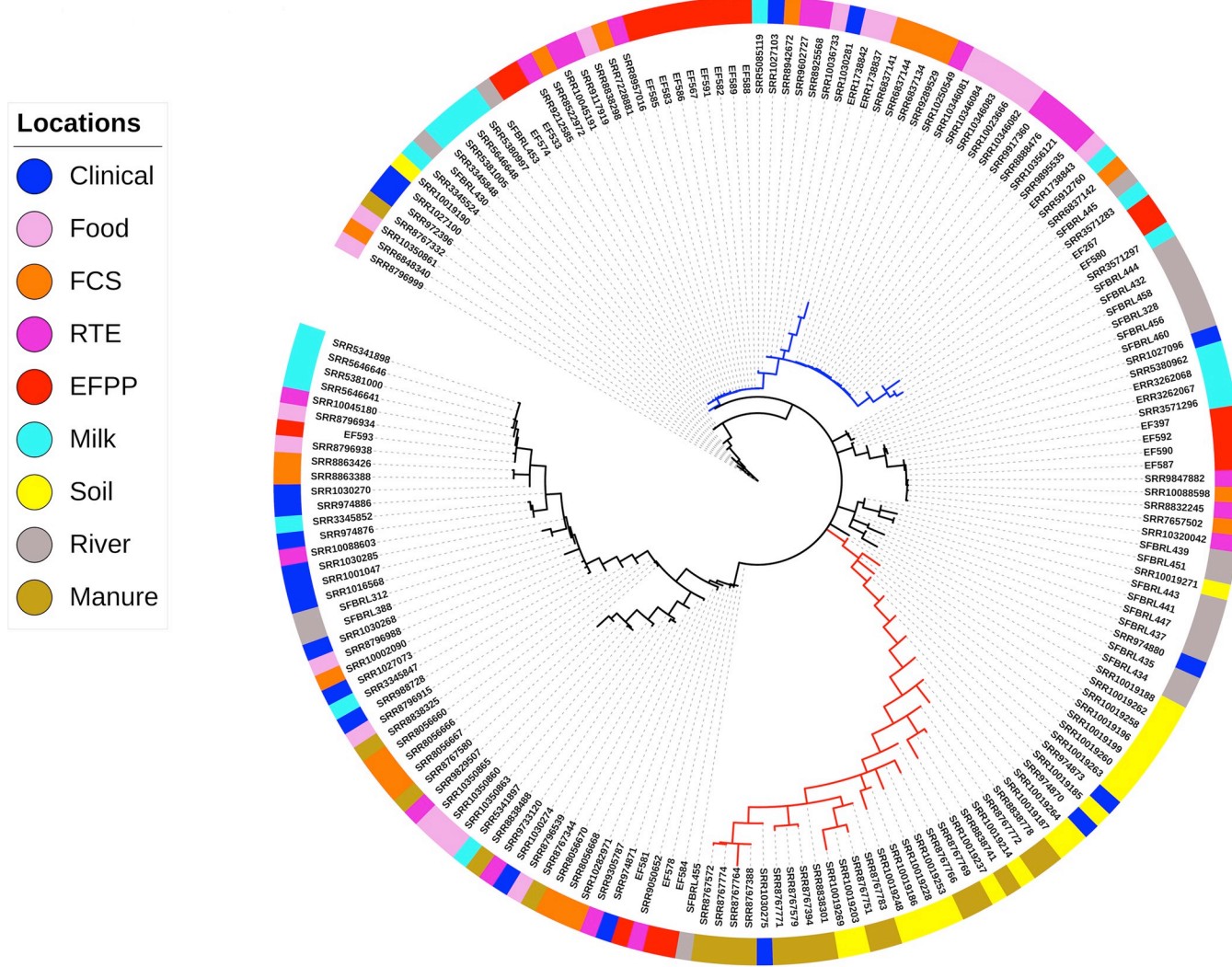

**Fig 1. Patterns of genetic differentiation in the 180 *L. monocytogenes* isolates.** Minimum spanning tree based on a distance matrix measuring pairwise differences in allelic content between isolate wgMLST profiles. The isolation source of each isolate is indicated with colors on the outer ring. Majority of the isolates sampled from soil and manure cluster together in a distant branch (red), suggesting their recent emergence from a common ancestor. A large number of food-related isolates cluster together in a single branch of the tree (blue), suggesting their close relatedness.

**Table 1. Pairwise genetic distances ($F_{ST}$) between groups of *L. monocytogenes* strains isolated from nine different ecological niches.**

|  | clinical | food | FCS | manure | milk | RTE product | soil | River water |
|---|---|---|---|---|---|---|---|---|
| **clinical** | 0 | | | | | | | |
| **food** | 0.051* | 0 | | | | | | |
| **FCS** | 0.062* | 0.015 | 0 | | | | | |
| **manure** | 0.067* | 0.126* | 0.137* | 0 | | | | |
| **milk** | 0.047* | 0.047* | 0.073* | 0.124* | 0 | | | |
| **RTE product** | 0.09* | 0.004 | 0.007 | 0.159* | 0.069* | 0 | | |
| **soil** | 0.064* | 0.11* | 0.124* | 0.019* | 0.104* | 0.135* | 0 | |
| **River water** | 0.094* | 0.091* | 0.107* | 0.153* | 0.069* | 0.092* | 0.113* | 0 |
| **EFPP** | 0.165* | 0.157* | 0.137* | 0.221* | 0.146* | 0.076* | 0.189* | 0.13* |

($^*$P < 0.05).

cluster. Secondly, a large number of food-related isolates (~51%, obtained from food, FCS, RTE products and EFPP) clustered together in a single branch of the tree (blue) with short branch-lengths to the tips, suggesting that these strains are closely related to each other. Although this is expected, it is interesting to find a few strains obtained from clinical cases (SRR1027103, SRR1030281), river water (SRR11051485, SRR11051480), and milk (SRR5085119, SRR5912760, SRR3571283, SRR3571297) in this cluster. The presence of isolates from unrelated ecological communities could be due to the technique used for constructing the dendrogram, which groups isolates based on pairwise differences in allelic content between isolate wgMLST profiles rather than characterizing differences between all variants in nucleotide sequences. For comparison with a reference strain of *L. monocytogenes*, the minimum spanning tree was rooted with EGD-e (S2 Fig).

The genetic differentiation test that computes pairwise $F_{ST}$'s between isolates collected from different ecological communities (Table 1) shows that isolates obtained from soil and manure show considerable genetic differentiation from isolates belonging to other communities, with the exception of isolates obtained from clinical cases. Secondly, isolates from the EFPP-RTE pairing has lower $F_{ST}$ than EFPP pairing from all other locations. Thirdly, the clustering dendrogram (Fig 1) and $F_{ST}$ test are supportive of each other in that isolates from RTE, FCS and food are not distinguished as separate populations.

We investigated LD between pairs of genes in the genome using an exact test, which measures non-random associations between alleles at two loci based on the difference between observed and expected allele frequencies. As expected, most genes pairs (~97%) in the genome of *L. monocytogenes* show significant LD among pairs of alleles (Fig 2, S4 File). A majority of genes (2205 of 2233, ~99%) were found to be at LD with at least 90% of other genes in the genome (S5 File). Of the remaining 27 genes (~1%) that were at LD with < 90% of genes (Table 2), 10 genes were found to be at LD with < 50% of genes. A single locus, *lmo0046*, was at LD with only 19 other genes.

## Discussion

Our dataset reveals the presence of strong LD in the genome of *L. monocytogenes*. Among the 2233 genes tested for LD, 2205 genes (approx. 99%) were found to have pairwise LD with a majority of other genes (90%) in the genome. High levels of LD can not only arise in highly clonal bacterial populations with low rates of recombination, but may also be temporarily present in bacteria with 'epidemic' population structures, in which high recombination rates randomize association between alleles, but adaptive clones emerge and diversify over the short-

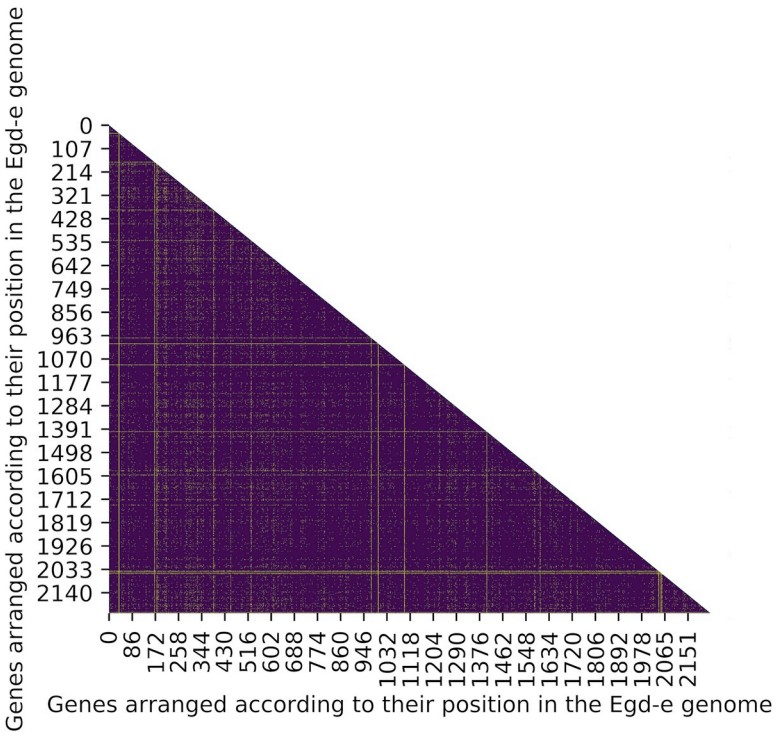

**Fig 2. Heatmap of the extent of LD in the genome of *L. monocytogenes*.** Genes are ordered according to their genomic positions in the *L. monocytogenes* reference strain EGD-e along the x and y axis (for gene names see S4 File). A majority of genes show significant LD in the genome (indigo), while few genes are at linkage equilibrium (yellow).

term [3, 5]. Because *Listeria* has a clonal genetic structure, it is unlikely that this high level of LD can arise except as a consequence of low rates of recombination. This is consistent with studies which report recombination in chromosomal genes as an infrequent event in natural populations of *L. monocytogenes* [8, 9]. Because the extent of genetic linkage is a useful index to the horizontal transfer occurring within a species and can be presented as direct evidence for recombination [3], the remaining ~1% of genes (Table 2) that were at LD with < 90% of genes can be described as "hot spots" for the gain of horizontally acquired information. The extensive linkage disequilibrium that we describe in *L. monocytogenes* is in sharp contrast to other pathogenic bacteria that are naturally competent for transformation and recombine frequently to give rise to either weakly clonal or panmictic population structures [35–37].

The *L. monocytogenes* pan-genome is highly conserved but open to limited acquisition of foreign DNA or genetic variability through evolutionary forces such as mutation, duplication or recombination [14]. Evidence for homologous recombination between closely related strains of *L. monocytogenes* has been detected by multiple studies, however, non-homologous recombination seems to be rare [12, 13, 38]. Although recombination via conjugation and generalized transduction has been reported in *L. monocytogenes* [39–41], and most competence related genes (which facilitate exogenous DNA uptake, for eg. comK, comE, comG etc.) are present in all *Listeria* genomes [42], natural competence or induced competence under laboratory conditions has not been observed in *L. monocytogenes* [43, 44]. This lack of competence may partially explain the low levels of gene acquisition from external gene pools. Limited gene acquisition may also be facilitated by defense systems for foreign DNA/mobile elements such as restriction-modification and/or CRISPR systems, both of which have been shown to restrict horizontal gene transfer in other bacterial genera [18].

**Table 2. Genes at LD with < 90% of genes in the genome of *L. monocytogenes*, showing significant evidence for horizontal genetic transfer.**

| Locus tag | Gene symbol | # Genes at LD | Percentage of genes at LD | Location in the chromosome (bp) | *Location in core/accessory genome w.r.t. BIGSdb-*Lm* | Function |
|---|---|---|---|---|---|---|
| lmo0046 | rpsR | 19 | 0.85 | 50514..50753 | core | small subunit ribosomal protein S18 |
| lmo2624 | rpmC | 185 | 8.289 | 2701254..2701445 | core | large subunit ribosomal protein L29 |
| lmo2856 | rpmH | 215 | 9.63 | 2943569..2943703 | accessory | large subunit ribosomal protein L34 |
| lmo1364 | cspL | 239 | 10.71 | 1387014..1387214 | accessory | Cold shock protein |
| lmo1469 | rpsU | 454 | 20.34 | 1501881..1502054 | core | small subunit ribosomal protein S21 |
| lmo2616 | rplR | 458 | 20.52 | 2697988..2698347 | accessory | large subunit ribosomal protein L18 |
| lmo1816 | rpmB | 484 | 21.69 | 1890951..1891139 | core | large subunit ribosomal protein L28 |
| lmo0248 | rplK | 576 | 25.81 | 265029..265454 | accessory | large subunit ribosomal protein L11 |
| lmo1335 | rpmG | 880 | 39.43 | 1363826..1363975 | core | large subunit ribosomal protein L33 |
| lmo0263 | inlH | 1006 | 45.07 | 284365..286011 | accessory | internalin H |
| lmo0582 | cwhA | 1223 | 54.79 | 618932..620380 | accessory | Invasion associated secreted endopeptidase |
| lmo2047 | rpmF | 1377 | 61.69 | 2130228..2130401 | accessory | large subunit ribosomal protein L32 |
| lmo2628 | rpsS | 1508 | 67.56 | 2702909..2703187 | accessory | small subunit ribosomal protein S19 |
| lmo2614 | rpmD | 1580 | 70.79 | 2697267..2697446 | core | large subunit ribosomal protein L30 |
| lmo0758 | - | 1606 | 71.95 | 783901..784788 | core | Hypothetical protein |
| lmo0514 | - | 1699 | 76.12 | 547520..549337 | accessory | Internalin |
| lmo0305 | - | 1709 | 76.57 | 329923..330999 | core | L-allo-threonine aldolase |
| lmo0659 | - | 1771 | 79.35 | 699410..700306 | accessory | Transcriptional regulator |
| lmo2206 | clpB | 1791 | 80.24 | 2294555..2297155 | accessory | Heat shock proteins |
| lmo0756 | - | 1797 | 80.51 | 781896..782801 | core | ABC Transporters |
| lmo0865 | - | 1859 | 83.29 | 903837..905510 | core | Amino sugar and nucleotide sugar metabolism |
| lmo2014 | - | 1888 | 84.59 | 2088797..2091454 | accessory | Glycan biosynthesis and metabolism |
| lmo1611 | - | 1904 | 85.3 | 1654902..1655975 | core | Aminopeptidase |
| lmo0264 | inlE | 1913 | 85.71 | 286219..287718 | accessory | Internalin E |
| lmo1839 | pyrP | 1925 | 86.25 | 1916166..1917452 | accessory | Electrochemical potential-driven transporters |
| lmo2179 | - | 1968 | 88.17 | 2264772..2268230 | accessory | Peptidoglycan binding protein |
| lmo0434 | inlB | 1981 | 88.75 | 457021..458913 | accessory | Internalin B |

*Location in core/accessory genome has been determined with respect to the core-genome MLST scheme developed by the Institut Pasteur [25].

The frequency of recombination in *L. monocytogenes* differs considerably in different regions of the genome and between isolates of different lineages [11, 19]. This may arise from differences in selective pressures in the environment and varying degrees of horizontal gene transfer. Several comparative genomic studies report a clustered distribution of accessory genes on the right replichore of the *L. monocytogenes* genome (approx. 500 Kb in the first 65˚),

indicating an area of high genome plasticity [14, 19]. On the contrary, a study by Orsi et al. failed to find any evidence of spatial clustering in a large number of genes which show evidence for recombination in *L. monocytogenes* [13]. Further, a recent study described the presence of homologous recombination in nearly 60% of loci in the core genome of *L. monocytogenes*, although most of this variation was also found to be affected by purifying selection and was thus neutral [25]. This is consistent with results from our analysis which finds linkage equilibrium between only ~1% of gene pairs in the genome. Also, genes considered as potential recombination hot spots (Table 2) in our dataset are found to be scattered in the genome. A large number (~41%) of these "hot spot" genes (*lmo0046*, *lmo2624*, *lmo2856*, *lmo1469*, *lmo2616*, *lmo1816*, *lmo0248*, *lmo1335*, *lmo2047*, *lmo2628*, *lmo2614*), encode ribosomal proteins and their related subunits. According to the complexity theory [45], informational genes involved in complex biosystems and maintenance of basal cellular functions are usually conserved, as they might be less likely to be compatible in the systems of other species. Thus, housekeeping genes such as ribosomal proteins are generally considered to be relatively restricted to horizontal gene transfer. However, several reports suggest horizontal gene transfer of ribosomal proteins in many prokaryotic genomes [46–49]. Two other "hot spot" genes (*lmo0865*, *lmo2014*) are involved in carbohydrate and amino acid metabolism and have shown evidence for recombination in a prior study [13], indicating that the rapid diversification of these genes may enable *L. monocytogenes* to adapt to environments with varying nutrient availabilities. Some of the other genes encode a variety of internalin's (*lmo0263*, *lmo0514*, *lmo0264*, lmo0434), transporters (*lmo0756*, *lmo1839*), transcriptional regulators (*lmo0659*), cell surface proteins (*lmo2179*), other invasion-associated proteins (*lmo0582*), and proteins involved in response to temperature fluctuations (*lmo1364*, *lmo2206*). Internalin's are cell surface proteins with known and hypothesized roles in virulence [18, 50]. Evidence of recombination in internalin's and these other genes suggests that *L. monocytogenes* is subjected to sustained selection pressures in the environment, and it responds to these pressures by continuously regulating its transcriptional machinery and remodeling the cell surface, thereby facilitating adaptation within the host and as a saprophyte.

In conclusion, we have identified the presence of strong linkage disequilibrium in the genome of *L. monocytogenes*. Parts of the genome showing strong non-random association between genes are highly conserved regions, and are most possibly affected by positive selection. The low levels of recombination within the *L. monocytogenes* genome suggests that the patterns of association observed between genes could be used to recognize newly emerging strains. As new strains are typed, their allelic configurations could be compared to other previously characterized strains. Novel allelic configurations would indicate a previously unobserved strain and can provide insights into the processes involved in the diversification and evolution of *L. monocytogenes*. Determination of evolutionary relationships between emergent strains and previously characterized pathogenic strains can help determine the potential of the emergent strain for causing disease. Such investigations can ultimately help to develop better control measures for this pathogenic microbe.

## Supporting information

**S1 File. Panel of 180 *L. monocytogenes* isolates collected from different ecological communities.**
(XLSX)

**S2 File. Whole-genome MLST profiles of the 180 *L. monocytogenes* isolates.**
(XLSX)

**S3 File. Whole-genome MLST profiles of 2233 loci retained for AMOVA after filtering out paralogous loci and loci with > 5% of missing data.**
(XLSX)

**S4 File. Heatmap of LD in the genome of *L. monocytogenes*.**
(XLSX)

**S5 File. Percentage of genes at LD with each gene in the genome of *L. monocytogenes*.**
(XLSX)

**S1 Fig. Workflow diagram for Haplo-ST.**
(PDF)

**S2 Fig. Minimum spanning tree of 180 *Listeria monocytogenes* isolates rooted with reference strain EGD-e.**
(PDF)

## Acknowledgments

We thank USDA and FSIS for providing us with *Listeria monocytogenes* whole-genome sequencing samples from river water and effluents of poultry processing plants. The high-performance computing cluster at Georgia Advanced Computing Resource Center (GACRC) at the University of Georgia provided computational infrastructure and technical support throughout the work.

## Author Contributions

**Conceptualization:** Richard J. Meinersmann.

**Data curation:** Swarnali Louha.

**Formal analysis:** Swarnali Louha.

**Funding acquisition:** Richard J. Meinersmann.

**Investigation:** Swarnali Louha.

**Methodology:** Swarnali Louha.

**Project administration:** Swarnali Louha.

**Resources:** Swarnali Louha, Richard J. Meinersmann.

**Software:** Swarnali Louha.

**Supervision:** Richard J. Meinersmann, Travis C. Glenn.

**Visualization:** Swarnali Louha.

**Writing – original draft:** Swarnali Louha.

**Writing – review & editing:** Swarnali Louha, Richard J. Meinersmann, Travis C. Glenn.

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
