## [Decision Letter · Decision Letter 0]

17 Dec 2020

PONE-D-20-33996

Whole genome genetic variation and linkage disequilibrium in a diverse collection of Listeria monocytogenes isolates

PLOS ONE

Dear Dr. Louha,

Thank you for submitting your manuscript to PLOS ONE. After careful consideration, we feel that it has merit but does not fully meet PLOS ONE’s publication criteria as it currently stands. Therefore, we invite you to submit a revised version of the manuscript that addresses the points raised during the review process.

Your manuscript has been reviewed by two experts in your field. Based on their comments, a major reviison is needed before a decision can be made.

We look forward to receiving your revised manuscript.

Kind regards,

Yung-Fu Chang

Academic Editor

PLOS ONE

Journal Requirements:

2. We note that you are reporting an analysis of a microarray, next-generation sequencing, or deep sequencing data set. PLOS requires that authors comply with field-specific standards for preparation, recording, and deposition of data in repositories appropriate to their field. Please upload these data to a stable, public repository (such as ArrayExpress, Gene Expression Omnibus (GEO), DNA Data Bank of Japan (DDBJ), NCBI GenBank, NCBI Sequence Read Archive, or EMBL Nucleotide Sequence Database (ENA)). In your revised cover letter, please provide the relevant accession numbers that may be used to access these data. For a full list of recommended repositories, see http://journals.plos.org/plosone/s/data-availability#loc-omics or http://journals.plos.org/plosone/s/data-availability#loc-sequencing.

Reviewers' comments:

Reviewer's Responses to Questions

**Comments to the Author**

1. Is the manuscript technically sound, and do the data support the conclusions?

Reviewer #1: Partly

Reviewer #2: Yes

2. Has the statistical analysis been performed appropriately and rigorously? 

Reviewer #1: Yes

Reviewer #2: Yes

3. Have the authors made all data underlying the findings in their manuscript fully available?

Reviewer #1: Yes

Reviewer #2: Yes

4. Is the manuscript presented in an intelligible fashion and written in standard English?

Reviewer #1: Yes

Reviewer #2: Yes

5. Review Comments to the Author

Reviewer #1: The manuscript by Louha et al. presented analysis of wgMLST data from 180 Listeria monocytogenes and evidence of linkage disequilibrium. The study is novel and has not been performed on L. monocytogenes, an important food-borne pathogen. The authors provided a thoughtful introduction and discussion of the utility of LD. Here are additional specific comments:

1. The references are not in the correct citation format within the text. The references should be numbered.

2. Line 82 – It is unclear whether the 180 L. monocytogenes were isolated by the authors, or that whole genome sequences data files were obtained for the analysis. Please clarify. If isolates were sequenced in-house, please provide protocols.

3. The Materials and Methods section should include the following information: a) isolation protocol for the 180 L. monocytogenes strains; b) DNA isolation method, library preparation, and sequencing; c) details of sequencing instrumentations, kit versions, sequencing data analysis software; and d) method of gene annotation and database used.

4. Line 127 – please provide reference for Fst ‘s “history of being used as a measure of the level of differentiation”.

5. Line 148 – please spell out “three”.

6. Line 148 and 152, - please list the SRA accession numbers for the isolates mentioned.

7. Figure 1 – Please add a reference L. monocytogenes (e.g. EGD-e) genome sequence to root the minimum spanning tree as a comparison.

8. Line 158 – Is the Figure 1 title supposed to be within the main text? It is unclear whether lines 158-164 are part of the Figure 1 title and description, or part of the main the text.

9. Line 170 – What is the “Secondly” and “Thirdly” on lines 169 and 170, respectively, following? I do not see a “First”, in this paragraph.

10. Line 191 – Please re-word the Table 2 title. A table title should not include “thus”.

11. Table 2 – please re-label the column “Gene name” to “Locus Tag”. Add another column for “Gene Symbol”. The gene symbol is the 3-4 letter abbreviation (e.g. lmo0046 gene symbol is rps R).

12. Line 200 – please provide reference for the statement “High levels of LD can not only arise in highly clonal bacterial populations with low rates of recombination”.

13. Line 204 – please clarify this statement without using the subjective word “difficult”. In addition, a lack of explanation does not provide convincing causal relationship between high level of LD and low rates of recombination. Please provide direct evidence for the cause.

14. Line 222 – please clarify what “competence related genes” are and provide examples of these genes.

15. Lines 267 – 172 – please explain further the notion of using LD in “identifying emerging strains of L. monocytogenes" and “developing better control measures”. The statement is general and not self-evident. Different DNA sequences in specific genes or LD ‘hot’ or ‘cold’ spots do not determine whether an isolate is a new strain. A comparison of whole genome sequences could identify new strains, and does not require the identification of LD. In addition, how would identification of LD lead to better ”control measures”?

16. Line 406 – S1 File – Please add SRA accession numbers to the isolates from River Water and Poultry Processing Plants

17. Figure 2 – Please provide X and Y axis labels.

Reviewer #2: The authors Louha, et al, of “Whole genome genetic variation and linkage disequilibrium in a diverse collection of Listeria monocytogenes isolates” examine whole genome sequence data from a set of L. monocytogenes from diverse sources to understand linkage disequilibrium and genetic variation in L. monocytogenes. The manuscript confirms existing knowledge in the field that L. monocytogenes has limited exchange of genetic information as a source for genetic variation. Overall the manuscript is well written and adds to our understanding of genetic variation in L. monocytogenes.

Major comments:

a. When listing panel of isolates in materials and methods – only list sources for 60 out of 180 isolates – would recommend listing sources for all isolates. How were the isolates selected that were included in this study if there were more than 20 isolates in the data set originally? Lineage, ST, and CC information should be provided for these isolates, as the authors note, there are differences between lineages and CC in terms of recombination. Also for clonal complex, important to understand if strains are of the same epidemic clone since that is also mentioned in the manuscript.

b. Line 96 – the authors used an unusual choice for assembly, the pipeline YASRA is not peer-reviewed (the reference given was from a dissertation), can the author provide further justification for using this assembler while there are new assemblers available that were build for Illumina data and longer paired end reads than the pipeline used in this manuscript.

c. Line 97 - The authors refer to the wgMLST scheme on the Institute Pasteur website for L. monocytogenes, a core genome MLST scheme is available there that has 1748 loci as part of the scheme, how are alleles for the remaining genes that are part of this manuscript’s wgMLST scheme called? Why are there 2 separate allele calling approaches? Overall this was very confusing and may benefit from a workflow diagram to be included in supplemental figures.

Minor comments:

a. Line 74 – could not find a Moura et al 2017 reference – please update

b. Line 260 – change “pleasures” to “pressures”

6. PLOS authors have the option to publish the peer review history of their article (what does this mean?). If published, this will include your full peer review and any attached files.

Reviewer #1: No

Reviewer #2: No

---

## [Author Response · Author response to Decision Letter 0]

29 Jan 2021

Reviewer #1: The manuscript by Louha et al. presented analysis of wgMLST data from 180 Listeria monocytogenes and evidence of linkage disequilibrium. The study is novel and has not been performed on L. monocytogenes, an important food-borne pathogen. The authors provided a thoughtful introduction and discussion of the utility of LD. Here are additional specific comments:

1. The references are not in the correct citation format within the text. The references should be numbered.

>>>We have numbered the references in the correct citation format within the text.

2. Line 82 – It is unclear whether the 180 L. monocytogenes were isolated by the authors, or that whole genome sequences data files were obtained for the analysis. Please clarify. If isolates were sequenced in-house, please provide protocols.

>>> Whole-genome sequencing data for 140 isolates from food, FCS, manure, milk, clinical cases, soil, and RTE products were obtained from the NCBI Pathogen Detection database. For the remaining 40 isolates from river water (#20) and poultry processing plants (#20), whole-genome sequencing data was provided to us by USDA and FSIS. These isolates were cultured and sequenced by USDA and the protocols have been provided in our recent paper (Louha et al. 2020, DOI: 10.1128/AEM.02248-20, citation no. 26 in manuscript). We have clarified this point and cited our paper for protocols in lines 77-83.

3. The Materials and Methods section should include the following information: a) isolation protocol for the 180 L. monocytogenes strains; b) DNA isolation method, library preparation, and sequencing; c) details of sequencing instrumentations, kit versions, sequencing data analysis software; and d) method of gene annotation and database used.

>>> We obtained whole-genome sequencing data for 140 isolates from the NCBI Pathogen detection database. The remaining 40 isolates were isolated and sequenced by USDA, and the protocols for that have been mentioned in detail in the paper “Louha et al. 2020, DOI: 10.1128/AEM.02248-20”. We have cited this paper (# 26) in the Materials and Methods section, line 83. 

Gene annotation i.e. wgMLST has been performed with the tool Haplo-ST, and the database used is BIGSdb-Lm (bundled with Haplo-ST). This has been described in details in lines 86-94 of the manuscript. We have also cited the paper (Louha et al. 2020, DOI: 10.1128/AEM.02248-20) which introduces and describes Haplo-ST in line 87. The online version of the BIGSdb-Lm database has been cited in line 94.

4. Line 127 – please provide reference for Fst ‘s “history of being used as a measure of the level of differentiation”.

>>> Two references (Siol et al. 2017, Bahbahani et al. 2018 [citation no. 32 and 33 in manuscript]) has been provided for “Fst ‘s “history of being used as a measure of the level of differentiation” in line 125. 

5. Line 148 – please spell out “three”.

>>> Changes have been made in line 145.

6. Line 148 and 152, - please list the SRA accession numbers for the isolates mentioned.

>>>SRA accession numbers for the isolates mentioned in line 145 and 150-151 has been listed in brackets (see highlighted text in these lines).

7. Figure 1 – Please add a reference L. monocytogenes (e.g. EGD-e) genome sequence to root the minimum spanning tree as a comparison.

>>>When the minimum spanning tree is rooted with EGD-e, its structure changes to some extent. While majority of the soil and manure isolates still cluster together to form the red branch, the food-related isolates (in the blue branch) cluster in a different pattern. Due to this change in tree structure, we have included the minimum spanning tree rooted with EGD-e in the Supplemental files as Figure S2 (lines 155-156).

8. Line 158 – Is the Figure 1 title supposed to be within the main text? It is unclear whether lines 158-164 are part of the Figure 1 title and description, or part of the main the text.

>>>PLOS ONE guidelines for figures mentions that:

“Figure captions must be inserted in the text of the manuscript, immediately following the paragraph in which the figure is first cited (read order). Do not include captions as part of the figure files themselves or submit them in a separate document.”

Hence we have inserted the Figure captions in the text, immediately following the paragraph in which the figure has been first cited. We have left a line gap before and after the Figure caption to separate it from the main text. For example, in the marked up manuscript, lines 158-164 are part of Figure 1 title and description.

9. Line 170 – What is the “Secondly” and “Thirdly” on lines 169 and 170, respectively, following? I do not see a “First”, in this paragraph.

>>>This paragraph lists three results apparent from the genetic differentiation test that computes pairwise FST’s. The “First” result is listed in the first line (line 166…169) of this paragraph.

10. Line 191 – Please re-word the Table 2 title. A table title should not include “thus”.

>>> Removed “thus” in the Table 2 title.

11. Table 2 – please re-label the column “Gene name” to “Locus Tag”. Add another column for “Gene Symbol”. The gene symbol is the 3-4 letter abbreviation (e.g. lmo0046 gene symbol is rps R).

>>> Changes have been made in Table 2 as suggested.

12. Line 200 – please provide reference for the statement “High levels of LD can not only arise in highly clonal bacterial populations with low rates of recombination”.

>>>The complete sentence in line 199-203 (“High levels of LD can not only arise in highly clonal bacterial populations with low rates of recombination, but may also be temporarily present in bacteria with ‘epidemic’ population structures, in which high recombination rates randomize association between alleles, but adaptive clones emerge and diversify over the short-term.”) has two references: Smith et al. 1993, and Feil and Spratt 2001 (citation no. 3 and 5). We have listed these two references at the end of the sentence in line 203.

Salmonella enterica can be a specific example of a species in which “high levels of LD arise in highly clonal bacterial populations with low rates of recombination” and has been mentioned in both the cited references.

13. Line 204 – please clarify this statement without using the subjective word “difficult”. In addition, a lack of explanation does not provide convincing causal relationship between high level of LD and low rates of recombination. Please provide direct evidence for the cause.

>>> We have replaced this statement in line 203-204 with “Because Listeria has a clonal genetic structure, it is unlikely that this high level of LD can arise except as a consequence of low rates of recombination.” Here we are not trying to establish a causal relationship, instead we are making a correlation between high levels of LD and low rates of recombination on the basis of the analysis that we have done. Similar correlations between high levels of LD and low rates of recombination has been made for other bacterial species in the literature (Smith et al. 1993, Feil and Spratt 2001 [citation no. 3 and 5]).

14. Line 222 – please clarify what “competence related genes” are and provide examples of these genes.

>>> Many species of bacteria are able to take up genetic material from their surroundings. Occasionally, such absorbed DNA is recombined into the organism’s own genome, resulting in natural transformation. Natural competence for transformation is considered a primary mode of horizontal gene transfer in prokaryotes, together with conjugation (direct cell to cell transfer of DNA via a specialized conjugal pilus) and phage transduction (DNA transfer mediated by viruses). Competence related genes are those that facilitate DNA uptake and consist of genes that encode the DNA uptake apparatus, the proteins that mediate protection of the incoming DNA within the bacterial cytoplasm, and the proteins that initiate recruitment of the recombination enzyme (Blokesch 2017). The DNA uptake machinery required for natural genetic competence is broadly conserved among species, including noncompetent bacteria. Examples of some competence related genes are comK, comG, comE etc. (Rabinovich et al. 2012).

We have described competence related genes and provided examples in line 219..”( which facilitate exogenous DNA uptake, for e.g. comK, come, comG etc.)”

References:

1) Blokesch M. Natural competence for transformation. Current Biology. 2016;26: R1119-R1136.

2) Rabinovich L, Sigal N, Borovok I, Nir-Paz R, Herskovits AA. Prophage excision activates Listeria competence genes that promote phagosomal escape and virulence. Cell. 2012;150: 792-802.

15. Lines 267 – 172 – please explain further the notion of using LD in “identifying emerging strains of L. monocytogenes" and “developing better control measures”. The statement is general and not self-evident. Different DNA sequences in specific genes or LD ‘hot’ or ‘cold’ spots do not determine whether an isolate is a new strain. A comparison of whole genome sequences could identify new strains, and does not require the identification of LD. In addition, how would identification of LD lead to better ”control measures”?

>>> We do not agree with “Different DNA sequences in specific genes or LD ‘hot’ or ‘cold’ spots do not determine whether an isolate is a new strain”. Due to recombination, emergent strains can have novel DNA sequences in their genes, thus giving rise to previously unobserved alleles (with no curated allele IDs present in databases). Thus, such newly arising strains would have novel allelic configurations (wgMLST profiles). Comparison of whole genome sequences can identify new strains, but this is cumbersome when compared to comparison of allelic profiles, which is the popular method currently used for identifying new strains by government laboratories (Jagadeesan et al. 2019).

Determination of LD in the genome helps recognize genes that are non-randomly associated (and thus less prone to recombination) and genes in which recombination is more likely. While recombination in the accessory genome (highly prone to horizontal gene transfer) of bacteria is an important source of evolutionary novelty, adaptation in highly conserved core genes is critical for long-term survival and short-term response to new selection pressures such as resistance to antibiotics (Everitt et al. 2014). As new strains are typed, their allelic configurations could be compared against other previously characterized strains. Novel allelic configurations (in both LD hot and cold spots) would indicate a previously unobserved emergent strain and can provide insights into the evolutionary changes and selection pressures in the environment that gave rise to the emergent strain. Determination of evolutionary relationship between emergent strains and known pathogenic strains can help determine the potential of the new emergent strain for causing disease. Thus, knowledge of the pathogenic potential of a newly arising strain and the selection pressures in the environment that led to the emergence of such a strain can help in developing better control measures for this pathogen. We have added this information in lines 263-268.

References:

1) Jagadeesan B, Baert L, Wiedmann M, Orsi RH. Comparative Analysis of Tools and Approaches for Source Tracking Listeria monocytogenes in a Food Facility Using Whole-Genome Sequence Data. Front Microbiol. 2019;10: 947.

2) Everitt RG, Didelot X, Batty EM, Miller RR, Knox K, Young BC et al. Mobile elements drive recombination hotspots in the core genome of Staphylococcus aureus. Nat Commun. 2014;5: 3956.

16. Line 406 – S1 File – Please add SRA accession numbers to the isolates from River Water and Poultry Processing Plants.

>>>SRA accession numbers for River Water and Poultry processing plant isolates have been added to File S1.

17. Figure 2 – Please provide X and Y axis labels.

>>>X and Y labels has been added to Figure 2.

Reviewer #2: The authors Louha, et al, of “Whole genome genetic variation and linkage disequilibrium in a diverse collection of Listeria monocytogenes isolates” examine whole genome sequence data from a set of L. monocytogenes from diverse sources to understand linkage disequilibrium and genetic variation in L. monocytogenes. The manuscript confirms existing knowledge in the field that L. monocytogenes has limited exchange of genetic information as a source for genetic variation. Overall the manuscript is well written and adds to our understanding of genetic variation in L. monocytogenes.

Major comments:

a. When listing panel of isolates in materials and methods – only list sources for 60 out of 180 isolates – would recommend listing sources for all isolates. How were the isolates selected that were included in this study if there were more than 20 isolates in the data set originally? Lineage, ST, and CC information should be provided for these isolates, as the authors note, there are differences between lineages and CC in terms of recombination. Also for clonal complex, important to understand if strains are of the same epidemic clone since that is also mentioned in the manuscript.

>>> Our dataset contains a total of 180 isolates. This consists of 20 isolates each from the following 9 locations: food, food contact surfaces (FCS), manure, milk, clinical cases, soil, ready-to-eat (RTE) products, river water, and poultry processing plants (20*9=180). Whole genome sequencing data for 140 isolates obtained from food, FCS, manure, milk, clinical cases, soil, and RTE products were obtained from the NCBI Pathogen Detection database. Whole genome sequencing data for the remaining 40 isolates obtained from river water and poultry processing plants were provided to us by USDA and FSIS. We have clarified this information in lines 77-83.

We have provided Isolation sources for all 180 isolates in File S1. We have also provided ST, CC and lineage of all isolates in File S1. Some isolates are new recombinant strains which have not been assigned ST, CC or lineage in the BIGSdb-Lm database, and hence this information has not been provided for these isolates. 

b. Line 96 – the authors used an unusual choice for assembly, the pipeline YASRA is not peer-reviewed (the reference given was from a dissertation), can the author provide further justification for using this assembler while there are new assemblers available that were build for Illumina data and longer paired end reads than the pipeline used in this manuscript.

>>>We used YASRA for assembly because YASRA is a comparative assembler which uses a template to guide the assembly of a closely related target sequence and can accommodate high rates of polymorphism between the template and target. Hence, this assembler can be used to assemble an allelic variant of a gene by mapping to a reference sequence, even when the target allele has diverged considerably from the reference gene sequence. We did test other newer assemblers built for Illumina data, but these assemblers failed to assemble alleles that were highly divergent from the reference template. We used YASRA as an assembler within the tool Haplo-ST, which was used for assembly and allele calling of our isolates. Although YASRA has not been peer-reviewed, Haplo-ST has been published recently (Louha et al. 2020, DOI: 10.1128/AEM.02248-20, citation no. 26 in manuscript). 

c. Line 97 - The authors refer to the wgMLST scheme on the Institute Pasteur website for L. monocytogenes, a core genome MLST scheme is available there that has 1748 loci as part of the scheme, how are alleles for the remaining genes that are part of this manuscript’s wgMLST scheme called? Why are there 2 separate allele calling approaches? Overall this was very confusing and may benefit from a workflow diagram to be included in supplemental figures.

>>>We have used only one approach (the tool Haplo-ST) to assemble and call alleles for L. monocytogenes isolates. We have mentioned this first in the Introduction, lines 68-73 and later modified the Materials and Methods, lines 86-94 to help resolve confusion.

Haplo-ST first cleans raw whole-genome sequencing reads using the FASTX-Toolkit, then it assembles alleles using YASRA, after which assembled alleles are assigned ID’s according to the nomenclature used by Institute Pasteur Listeria monocytogenes database using BIGSdb (tool used for calling alleles on the Institut Pasteur website). This pipeline has been automated with scripts and made portable by installation of all software dependencies i.e. FASTX, YASRA, and BIGSdb-Lm within a local Linux Virtual machine. A workflow diagram for Haplo-ST has been included in the paper (Louha et al. 2020, DOI: 10.1128/AEM.02248-20) that describes it. We have cited this paper in line 94 (citation no. 26) and also included the workflow diagram for Haplo-ST in Supplemental Fig S1.

We have used a wgMLST scheme consisting of 2554 genes for allele calling (mentioned in line 94). This scheme includes the 1748 loci in the core genome MLST scheme on the Institute Pasteur website as well as other genes present in the Institute Pasteur database. This has been described in details in the paper cited for the tool Haplo-ST.

Minor comments:

a. Line 74 – could not find a Moura et al 2017 reference – please update

>>> Thanks for pointing this out. We have changed the reference from ‘Moura et al 2017’ to ‘Moura et al 2016’, in line 68, citation no. 25. 

b. Line 260 – change “pleasures” to “pressures”

>>> Necessary changes have been made in line 255.

---

## [Decision Letter · Decision Letter 1]

12 Feb 2021

Whole genome genetic variation and linkage disequilibrium in a diverse collection of Listeria monocytogenes isolates

PONE-D-20-33996R1

Dear Dr. Louha,

We’re pleased to inform you that your manuscript has been judged scientifically suitable for publication and will be formally accepted for publication once it meets all outstanding technical requirements.

Kind regards,

Yung-Fu Chang

Academic Editor

PLOS ONE

Additional Editor Comments (optional):

Reviewers' comments:

Reviewer's Responses to Questions

**Comments to the Author**

1. If the authors have adequately addressed your comments raised in a previous round of review and you feel that this manuscript is now acceptable for publication, you may indicate that here to bypass the “Comments to the Author” section, enter your conflict of interest statement in the “Confidential to Editor” section, and submit your "Accept" recommendation.

Reviewer #2: All comments have been addressed

2. Is the manuscript technically sound, and do the data support the conclusions?

Reviewer #2: Yes

3. Has the statistical analysis been performed appropriately and rigorously? 

Reviewer #2: Yes

4. Have the authors made all data underlying the findings in their manuscript fully available?

Reviewer #2: (No Response)

5. Is the manuscript presented in an intelligible fashion and written in standard English?

Reviewer #2: Yes

6. Review Comments to the Author

Reviewer #2: (No Response)

7. PLOS authors have the option to publish the peer review history of their article (what does this mean?). If published, this will include your full peer review and any attached files.

Reviewer #2: No

---

## [Editor Report · Acceptance letter]

17 Feb 2021

PONE-D-20-33996R1 

Whole genome genetic variation and linkage disequilibrium in a diverse collection of *Listeria monocytogenes* isolates 

Dear Dr. Louha:

I'm pleased to inform you that your manuscript has been deemed suitable for publication in PLOS ONE. Congratulations! Your manuscript is now with our production department. 

Kind regards, 

on behalf of

Dr. Yung-Fu Chang 

Academic Editor

PLOS ONE